



# Magnetostatic reciprocity for MR magnet design

Pedro Freire Silva, Mazin Jouda, and Jan G. Korvink

Karlsruhe Institute of Technology (KIT), Institute of Microstructure Technology, Karlsruhe 76131, Germany

**Correspondence:** Jan G. Korvink (jan.korvink@kit.edu)

**Abstract.** Electromagnetic reciprocity has long been a staple in MR radio-frequency development, offering geometrical insights and a figure of merit for various resonator designs. In a similar manner, we use magnetostatic reciprocity to compute manufacturable solutions of complex magnet geometries, by establishing a quantitative metric for the placement and subsequent orientation of discrete pieces of permanent magnetic material. Based on magnetostatic theory and nonlinear FEM simulations, it is shown how assembled permanent magnet setups perform in the embodiment of a variety of designs, and how magnetostatic reciprocity is leveraged in the presence of difficulties associated with self-interactions, to fulfil various design objectives, including self-assembled micro magnets, adjustable magnetic arrays, and an unbounded magnetic field intensity in a small volume, despite realistic saturation field strengths.

## 1 Introduction

Magnetic field generation is an intrinsic requirement of magnetic resonance (MR) equipment, providing both the source of Zeeman polarisation, and the precession field. Despite a sharp dependence of the MR sensitivity on field intensity, the high cost of superconducting magnets has resulted in an increase in the use and development of compact permanent-magnet systems, starting in the 1980's (Zakotnik and Tudor (2015)) with the discovery of high-remanence and high-coercivity magnetic materials such as Neodymium-based alloys. As better detection methods, hyperpolarization techniques, and excitation schemes lower limits of detection, new applications based on these magnet systems have been made possible by offsetting the SNR loss stemming from the reduced magnetic field intensity. Beyond a cost reduction, permanent magnets have allowed for portable and non-standard MR applications and can be divided into narrow-bore spectroscopy magnets (Danieli et al. (2010); Chonlathep et al. (2017)), large-bore imaging magnets (Hugon et al. (2010); Soltner and Blümler (2010); Windt et al. (2011); Cooley et al. (2018)), and other speciality magnets providing an external homogeneous volume (Utsuzawa and Fukushima (2017); Paulsen et al. (2008); Chang et al. (2011); Marble et al. (2007); Manz et al. (2006)), profiling gradients (Rahmatallah et al. (2005); Landeghem et al. (2012); Chang et al. (2006); Marble et al. (2006); Luo et al. (2018)), or pre-polarisation fields (Tayler and Sakellariou (2017); Raich and Blümler (2004)).

Despite the numerous published concepts, several key bottlenecks and cost factors remain, hindering the design and construction of suitably good quality magnet systems. Reported challenges include the complexity in using tens of source magnets (Hugon et al. (2010); Soltner and Blümler (2010); Raich and Blümler (2004)), magnetisation strength error (Hugon et al. (2010); Soltner and Blümler (2010); Raich and Blümler (2004)) of 1%, magnetisation direction error (Hugon et al. (2010)) of





2°, magnet size error (Raich and Blümler (2004)) of 0.1 mm, as well as experimental difficulties regarding assembly-alignment errors and the large forces associated with strong permanent magnets. As target inhomogeneities in MR lie below the parts-per-million (ppm) range, *a priori* magnet deviations of 1% constitute a massive obstacle in achieving high homogeneity magnets.

Current methodology requires, for example, the time-consuming analysis of each magnet in a large pool, so as to use only the most similar magnets (Hugon et al. (2010)), the individual characterisation and ordering of an array to reduce inhomogeneity (Hugon et al. (2010); Soltner and Blümler (2010); Cooley et al. (2018)), or the use of high-precision mechanical adjustment systems (Danieli et al. (2010)). All of these approaches limit the number of magnets in a system, and as a result, the final field homogeneity. These efforts are often not possible or sufficient, and thus additional passive shimming techniques

are required (Hugon et al. (2010)), in addition to active shimming.

So as to overcome these limitations, and building upon the self-assembly idea mentioned in Chandrana et al. (2015), the long-known principle of magnetostatic reciprocity, also known as Betti's theorem (Insinga et al. (2016b); Mikhlin (1964); Betti (1872)), was considered for generalised Halbach-like MR permanent magnet design. Geometrical reciprocity is ever-present in MR in the form of electromagnetic reciprocity from Hoult and Richards (1976), which creates a correspondence between

a current distribution in the MR detection coil and the signal induced in it. Similarly, one can deduce a reciprocity between two regions in space and the magnets/fields contained therein. This allows for the quantitative *evaluation* of proposed magnet designs, as well as in planning the *assembly* of the geometry. An easier alignment/assembly allows for the use of a larger number of discrete magnets, which in turn allows for the discretisation of magnetisation distributions with better fidelity, and the use of smaller, cheaper and safer magnets (Zakotnik and Tudor (2015)).

The principle and its direct and indirect consequences are initially discussed in Sect. 2.1.1, subsequently enabling the computation of a material-independent figure of merit for magnetic systems in Sect. 2.1.2. In a similar analytical approach, the impact of an increasing number of magnets on the field quality of a Halbach array was researched, in Sect. 2.1.3, to showcase the benefits of being able to handle a larger number of magnets.

To demonstrate the breadth of possibilities and insights allowed when designing with the principle, three different applica-

tions were proposed. The first leverages the minimum-energy state that comes with maximum magnetic coupling, for an easier auto-assembly of a Halbach magnet, and is shown in Sect. 2.2.1. A second application proposes an arbitrarily chosen development goal and shows how the reciprocity principle can elucidate the design process on each step, allowing for the complex behaviour shown in Sect. 2.2.2. The final application maximally leverages the principle, enabling a topological optimisation method to obtain the highest possible magnetic field in a single-sided magnet, which shows an unbound magnetic field strength

in Sect. 2.2.3. The methods used for simulation, material modelling and result post-processing are shown separately in Sect. 3, for clarity, and the paper concludes with a discussion of the results and an outlook for the technique in Sect. 4.





## 2 Results

### 2.1 Theoretical insights

#### 2.1.1 Principle corollaries

We consider a region of space with defined magnetic vector potential $A$, which generates a magnetic field $H_A$ in the surrounding space, and a second region with a predefined magnetisation $M$, which also generates a magnetic field $H_M$ in its surrounding space. The energy reciprocity between the two specified fields can be stated as per Jeans (1908):

$$\int H_M \cdot M_A \, d\mathbb{R}^3 = \int H_A \cdot M_M \, d\mathbb{R}^3 \tag{1}$$

In other words, the magnetic vector potential's equivalent magnetic moment $M_A$ has an energy in the magnetic moment's
external field $H_M$ that is equal to the energy of the original magnetic moment $M_M$ in the external field of the vector potential $H_A$. If one assumes high-coercivity magnets with an effective tensorial permeability $\mu_0\mu_r$ and remanent field $B_r$, i.e., a model which approximates well the behaviour of common hard magnetic materials such as NdFeB compounds, it is possible to set $M = (\mu_r - 1)H + \mu_0^{-1}B_r$. This simplifies Eq. 1 to:

$$\int B_{r_A} \cdot H_M \, dV_A = \int B_{r_M} \cdot H_A \, dV_M$$

$$+ \left( \int [\mu_0(\mu_{r_M} - 1) \cdot (H_A + H_M)] \cdot H_A \, dV_M \ - \int [\mu_0(\mu_{r_A} - 1) \cdot (H_A + H_M)] \cdot H_M \, dV_A \right) \tag{2}$$

Whereas this approach matches exactly the ideal coupling between two equal magnets, this remains an approximation when developing MR applications, as the permeability of samples and magnets differs. One can nonetheless estimate the relative inefficiency of a design in optimally placing magnetic energy in the anchor volume in which a sample will be placed. This inefficiency is dominated by the relative amplitude of the last two terms of Eq. 2 w.r.t. the first one on the right. Beyond being a differential term with a similar integrand and integration volume, the last two terms are further multiplied by a reductive coef-
ficient, $(\mu_r - 1)$, which takes small results in neodymium magnets as $\mu_\parallel = 1.03$ and $\mu_\perp = 1.12$ (Katter (2005)), for example. This clearly shows how the first term on the right remains dominant, energy-wise, and one can thus approximate by a complete linear superposition.

Equation 1 holds for an arbitrarily small $M$, which indicates that a higher anchor field in a location will also mean a larger contribution to the target field from magnetisation located there. Similarly, the discussed method does not elucidate the
boundary design which leads to an homogeneous target field.

Assuming $B_{target} = -B_{r_A}$ in Eq. 1, one can see how maximum compliance with the target field is given by the minimum magnetostatic energy condition. This corollary indicates that a freely rotating magnet will tend to the direction that better generates the target field needed to minimise the energy with the anchor magnet, thus providing the force needed for *self-assembly*.

Just as this effect proves the hypothesis stated in Chandrana et al. (2015), used to auto-assemble 8 magnets into a Halbach (Halbach (1980)) array, it also elucidates an important limitation that arises, namely that of the self-interaction of the





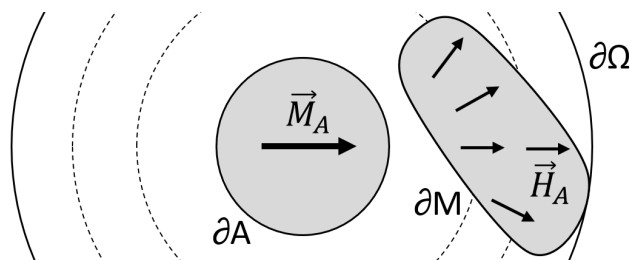

**Figure 1.** Representation of the domains involved in the reciprocity principle. As an example, the magnet under development, outlined by the boundary $\partial M$ of $\Omega_M$, is to create a homogeneous field across the anchor magnet volume outlined by the boundary $\partial A$ of $\Omega_A$. Reciprocally, the field created by anchor magnet $\boldsymbol{x}_A \in \Omega_A$ on the magnet volume $\boldsymbol{x}_M \in \Omega_M$ is shown and defines the direction that would better create a field in $A$ with a uniform direction, i.e., reciprocity implicitly specifies a suitably matched Halbach distribution.

designed magnetisation distribution. The ideal magnetisation distribution has a minimum energy in the interaction with the anchor magnet, but not on its self-interaction energy, which has it evolving to a global energy minimum. For this reason, each individual magnet must be aligned individually, or the self-interaction must be countered with secondary magnetic fields, so as

to have a high fidelity reproduction of the ideal distribution.

### 2.1.2 Coupling parameter

Knowing the remanent field distribution that maximises the magnetic field in one volume, one can thus compute the efficiency of an array in matching that distribution. In the case shown, the volume that can be filled with magnet material is contained between $\partial A$ and $\partial \Omega$, corresponding to the domain $\Omega$. The outer boundary is defined as the weakest contour line of $H_A$ that

intersects $M$, as indicated in Fig. 1, but this definition can be adapted depending on the application and may be replaced by a symmetry line (*e.g.* profilometry applications can only populate a half-space). On this volume, one can compute the coupling parameter $\eta$, which computes how well the magnet under design, defined by its normalised remanent magnetisation $b_r = B_r/max(|B_r|)$, is aligned with the field created by the anchor, as per:

$$\eta = \overline{\overrightarrow{H_A} \cdot \overrightarrow{b_r}} = \frac{\int (\overrightarrow{H_A} \cdot \overrightarrow{b_r}) \, dV_M}{\int |\overrightarrow{H_A}| \, dV_\Omega} \tag{3}$$

For example, in an ideal 2D Halbach magnet with $\eta = 1$, the target field has perfectly circular contour lines, so that $\Omega$ is bounded by the least contributing magnetisation, the outer diameter of the magnet array, and the boundary of the circular sample, the inner diameter of the magnet array. These match exactly the placement and direction of magnets in an ideal 2D Halbach array and the coupling is thus optimal. Real implementations with sub-optimal filling factors or supporting structures will however have $\eta < 1$. The coupling parameter compares designs quantitatively and provides a computable cost to inefficient

but necessary design techniques, like introducing a dead-volume to increase homogeneity (Hugon et al. (2010); Soltner and Blümler (2010)), using an anti-aligned piece to cancel first order field gradients (Manz et al. (2006); Perlo et al. (2006)), or overextending a magnet to emulate an infinite height (Paulsen et al. (2008)).





### 2.1.3 High-*n* applications

A design choice that often directly impacts the performance and complexity of a magnetic arrangement is the number of
individual parts involved. A large number of these will self-evidently allow for a better discretisation of a desired magnetisation
profile. This introduces the possibility of a stronger magnetic field, by better matching the functional introduced in Eq. 3, but
the dependence on the experimental effects governing field inhomogeneity remains unclear.

For high homogeneity applications using permanent magnets, the centre of a 2D discrete Halbach field remains the ideal
placement area for a test sample, due to a saddle point of the magnetic field intensity in the axis of the arrangement. A sym-
metrical placement of four magnets, or a combination thereof, will show a saddle point with zero first and second derivatives
of the field intensity. This means that any deviation from an ideally-built discrete array can be modelled well, locally, by the
total effect of each contribution on its directional, first-order derivative at the origin.

Consider the magnetic field generated by one 2D cylindrical magnet of radius $r_0$ and uniform magnetisation $M_0$, as per Jeans
(1908), with an angle defined by a rotation matrix $R_\theta$, and placed at $\{x_0, y_0\}$:

$$B(x, y, x_0, y_0, \theta) = \frac{M_0 r_0^2 \mu R_\theta \cdot \{(x - x_0)^2 - (y - y_0)^2), 2(x - x_0)(y - y_0)\}}{((x - x_0)^2 + (y - y_0)^2)^2} \tag{4}$$

Summing the effect of various cylinders allows for the analytical description of the field generated by an array of $n$ magnets in
a Halbach configuration, placed on the unitary circle:

$$B_H(x, y) = \sum_{i=1}^{n} B(x, y, cos(2\pi i/n), sin(2\pi i/n), -4\pi i/n) \tag{5}$$

This equation enables the computation of $\nabla |B_H|.v$ at the origin, for an increasing number of magnets. Assuming that the
experimental uncertainty in magnetisation intensity, angular direction and piece dimensions have a small relative deviation, $\epsilon_i$,
one can use the small angle approximation and obtain the following figures of merit (FOM):

$$\text{FOM}_{\Delta Mag} \propto \text{FOM}_{\Delta r_0} \propto \nabla | \sum_{i=1}^{n} (1 + \epsilon_i) B_i | \cdot v = \sum_{i=1}^{n} c_i(v) \epsilon_i \tag{6}$$

$$\text{FOM}_{\Delta\theta} \approx \nabla | \sum_{i=1}^{n} \left( \begin{smallmatrix} 1 & -\epsilon_i \\ \epsilon_i & 1 \end{smallmatrix} \right) B_i | \cdot v \propto \sum_{i=1}^{n} c_i(v) \epsilon_i + n^{-1} \sum_{i,j=1}^{n} k_{ij}(v) \epsilon_i \epsilon_j \tag{7}$$

In Eq. 6 and 7 one arrives at a total inhomogeneity that is dominated by a weighed linear sum of the various effects. Assuming
that $\epsilon_i$ has a Gaussian distribution around a zero mean, it becomes clear that the field inhomogeneity will increase as $\sqrt{n}$ as the
the sum of statistically independent distributions of the magnet's behaviour.

Due to a linear superposition, the field scales with the number of magnets, $n$, while the effects dominating the inhomo-
geneity of the field, originating from fabrication errors, scale as $\sqrt{n}$, which means an overall improvement of the relative field
inhomogeneity of $1/\sqrt{n}$. This shows a deep incentive in increasing the total amount of magnets in an assembly, which follows
common fabrication approaches (Hugon et al. (2010); Soltner and Blümler (2010); Raich and Blümler (2004)).



## 2.2 Applied reciprocity

### 2.2.1 Auto-alignment

An impactful application of the reciprocity principle is the automatic assembly targeting micro-scale applications, which, due
to the small dimensions and the large relative forces involved, make assembly especially challenging. To best showcase the
possible approaches, a micro-array and a powder magnet were simulated in their self-alignment and corrected alignment,
attained through application-dependent correction pieces.

A Halbach micro-array was initially conceived to be deposited as a single material layer on a substrate, uniformly magne-
tised, and allowed to self-align. This enables the use of a low fringe-field design, with a strong field intensity, in integrated
applications, while using any desired magnetic material. A quarter of the array was simulated as a 2D model with sizes
ranging up to the maximum packing of 8 circles in the available space $\Omega$, between the normalised inner radius and $d_M$.
The results in Fig. 2 clearly show the impact of outer radius $d_M$ on field strength and homogeneity for one 8-magnet ring,
which is further exacerbated as several magnet rings are employed. An improvement of two orders of magnitude is seen on the
homogeneity, as first order field gradients cancel out in the correct alignment, thus showing that correction is a critical step.
From Fig. 2A, it is clear that the procedure uses only indirect handling of the magnets and requires only sequential liberation
and fixation of magnets. The resulting ease of fabrication and the zero scaling law of permanent magnets make these ideal for
down-scaling magnetic arrays to a length-scale far below what is currently achievable for complex arrays.

A natural follow-up to the results shown is the evolution from a finite number of well-defined magnetic structures to arbi-
trarily many, much like in a powder magnet. These allow for easy tooling, the drastic reduction of statistical variance errors as
seen in Sect. 2.1.3 (by shifting these to fabrication precision), and large height micro-magnets, overcoming sputtering height
limitations. A powder can reach a maximal theoretical sphere packing density of 74% in an FCC/HCP lattice but, without
correct placement, will reach a random packing after vibration annealing, of 64% (Jaeger and Nagel (1992)). A mix of pow-
ders of staggered sizes can however reach arbitrarily close to a 100% packing factor, as smaller powders fill the gaps left
by larger pieces, or even emulate a single phase with composite effects. Using two powder species with different tempera-
ture coefficients of remanence, $\alpha$, one positive and one negative, it is possible to locally cancel temperature-induced drifts
in field strength or even offset volume drifts by tuning the expansion of the binding agent. A thermally-compensated mix of
NdFeB ($B_R = 1.45T, \alpha \approx -0.12\%/K$) and SmEr ($B_R = 0.89T, \alpha \approx +0.11\%/K$ in Chen et al. (2002)), for example, shows
an effective packing density of 75% when compared to NdFeB alone.

Figure 3 shows that the auto-alignment of a powder structure is possible and able to provide a significant field intensity,
while leveraging all of the advantages mentioned above. Field intensity is seen to level-off as the self-interaction of the powder
dominates over the alignment field of the anchor, which indicates a need for the use of correction structures. This is further
emphasised by the large inhomogeneity gap between the self-aligned and the Halbach conditions. Due to the complexity of the
endeavour and its application-specific nature, further investigation of this effect was deemed to be outside of the scope of the
present manuscript. As a baseline, the ideal Halbach alignment was also simulated, which explicitly shows the critical effect
of disregarding the marginal susceptibility in hard magnetic materials.



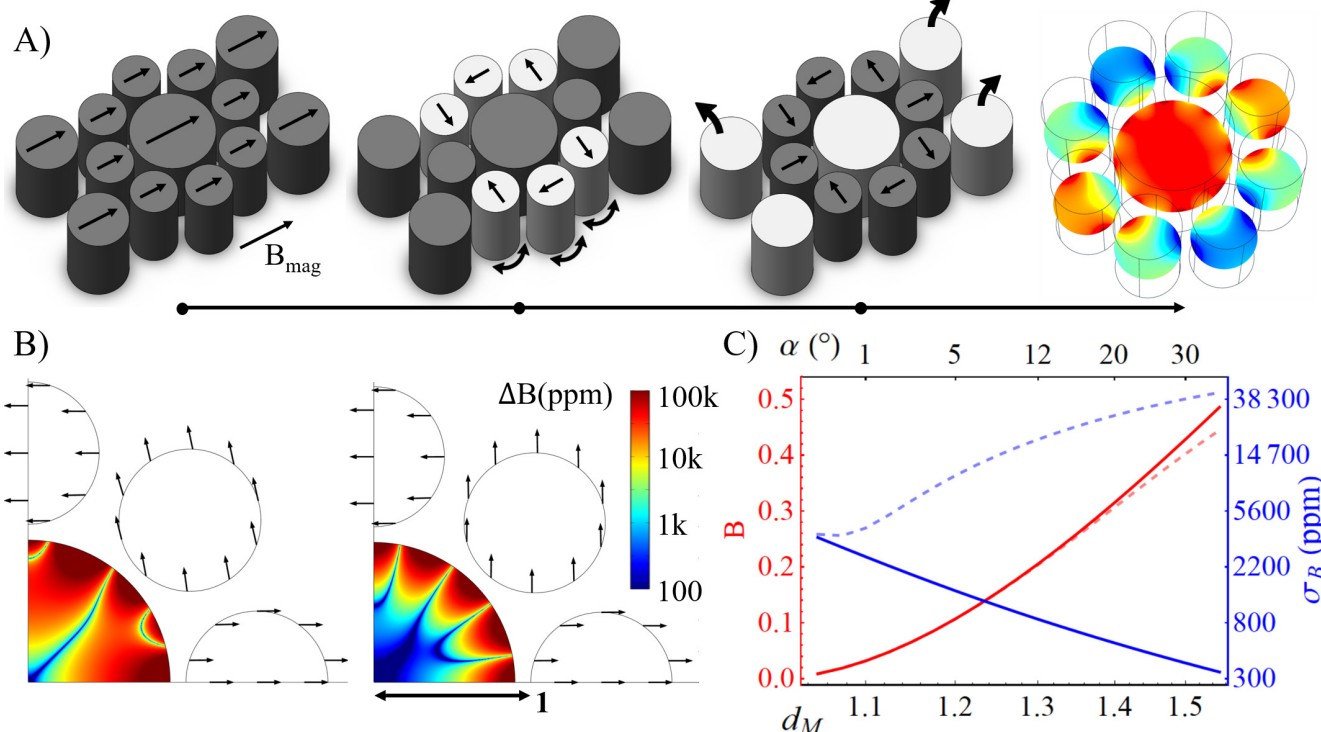

**Figure 2.** A) The alignment process. Ferromagnetic thin film structures are deposited and magnetised uniformly along the lateral direction. Next, non-aligned structures are released from the substrate and allowed to rotate to their force equilibrium positions, after which they are fixed again to the substrate. Finally, correction structures are released and lifted off, leaving the final field distribution. Structures active in a step are shown brighter, and their magnetisation directions are indicated by straight arrows. Curved arrows indicate movement. B) and C) The homogeneity of the arrangement in A) without and with the outer alignment structures, respectively. The line plots in C) reveal the standard deviation of the magnetic field, $\sigma_B$ (i.e. the homogeneity), as a function of the radial distance normalised to the anchor's radius, $d_m$, and the tilt angle $\alpha$, for the self- (dashed lines) and corrected (full lines) alignments.

Whereas a full theoretical Halbach possesses zero inhomogeneity, the real response of the magnetic material to the demagnetising field significantly alters this expectation to the behaviour, as shown in Fig. 3B.

### 2.2.2 Adjustable operation

One of the advantages of an auto-assembled magnet is the ease at which it can be put together, usually a secondary concern as
homogeneous MR magnets are only assembled once and then used in a static assembly. Similarly, MR profiling magnets are *set* for a specific working condition and thus field strength, discernible slice thickness, and penetration depth become magnet parameters that cannot be subsequently changed. However, with an auto-adjusting application one could dynamically tune the operational conditions as needed, making use of the coupling force introduced by a tuning piece.



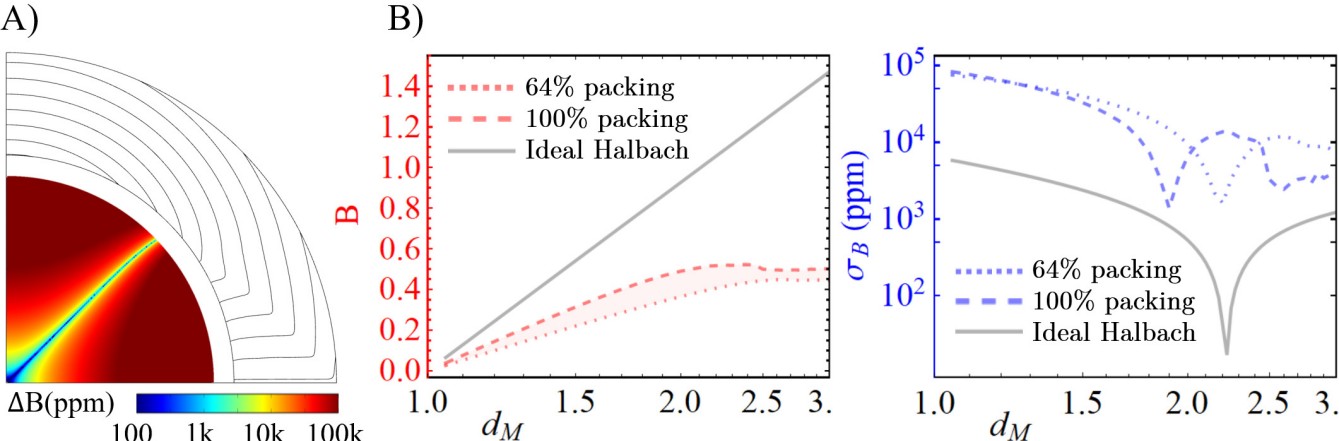

**Figure 3.** A) shows the implementation of the structures in Fig. 2, in their self-aligned, minimum-energy condition, from powdered material. B) Dependence of field intensity and homogeneity, inside the inner half-radius, on an increasing magnet size, as defined by the outer radius normalised by the anchor's radius, $d_m$.

By solving the inverse problem, the reciprocity principle readily facilitates application development and the interpretation of results. Setting out to generate an adjustable profiling magnet with a large penetration depth, an anchor was set at different heights above a plane below which magnets could be placed. The variation of the field direction on the design region, as the anchor is moved, reciprocally indicates the variation the magnets need to have to adjust to a target field at different penetration depths, as illustrated in Fig. 4A with rotating cylinders. This rotation can then be induced mechanically or with a magnetic control piece. Once more leveraging the principle, the contribution of each point in space to the field on the anchor space and on the cylinder space can be computed. With these, one can optimally find the region in space which strongly interacts with the cylindrical array, inducing torque, but has a minimum contribution to the target field, reducing any adverse effects. These pieces were implemented as cubes, for simplicity, as seen in Fig. 4A.

The application requires a sectional symmetry to generate the nearly axially symmetric field which allows for a minimal discernible slice. As the alignment of all magnets at the same rotation angle is a meta-stable configuration of the cylinder-array, a scaffold is needed to maintain the relative angular positions. A stable configuration of the coherent cylinder-array is a parallel orientation pointing upwards/downwards and, due to the desired high packing density, the strong interaction between the cylinder-array elements makes it hard to shift the configuration away from its minimum energy alignment. For this reason, a set of control pieces with identical upward magnetisation is place between the polarising magnets, creating a local energy minimum somewhat shallower than the absolute minimum, which allows for adjusting the strongly aligned magnet array with smaller pieces. The figures of merit of the profiling magnet assembly are its field strength, its penetration depth, and the discernible slice thickness. The minimum thickness comes from the condition $T\gamma\nabla_z B > \Delta\omega_{\text{slice}}$ and was implemented, for the field in a cylindrical slice of radius R, as:

$$T_{\min}(R,z) = \frac{\text{Max}(\Delta|B|)|_{r<R}}{\text{Min}(\nabla_z|B|)|_{r<R}} \tag{8}$$

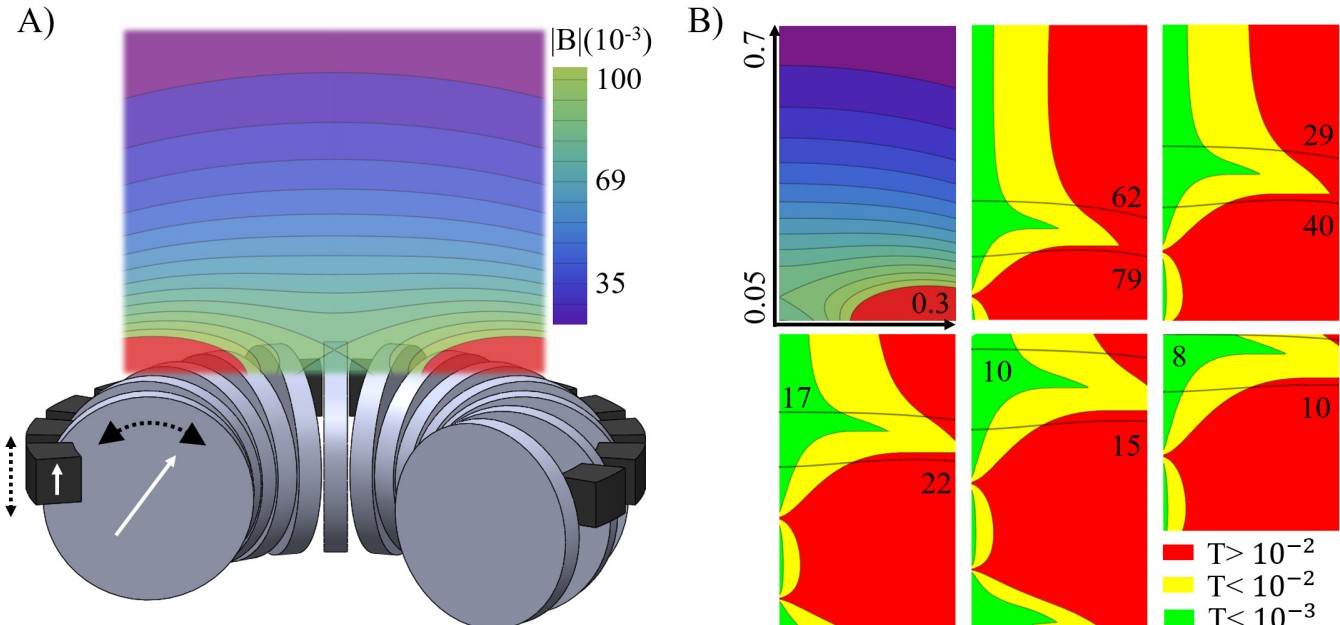

**Figure 4.** A) Section of a variable magnet geometry, with a toroidal array of disk magnets in grey, magnet control pieces in black, and the resulting magnetisation directions as white arrows. The resulting magnetic field intensity contours are shown at the minimum energy configuration and feature a uniform vertical gradient region above the magnet array. B) The top left and centre plots show the magnetic field intensity contours and minimum discernible thickness for the magnet configuration in A) as defined by Eq. 8. B) The profiles arising from actuation of the control pieces for $z \in \{-0.06, 0.08\}$, rotating the cylinders by $\theta \in \{-32°, 27°\}$. The normalised magnetic field intensity, in normalised milliunits, and its contour lines, are shown in black.

**Table 1.** Normalised figures of merit, as described in Sect. 3, comparing the obtained results to literature.

|  | d (%) | B | $\nabla_z |B|$ | T ($10^{-3}$) |
|---|---|---|---|---|
| Figure 4 | 25-68 | 0.008-0.07 | 0.2-0.3 | <1 |
| Rahmatallah et al. (2005) | 25 | 0.18 | 0.36 | 0.7 |
| Landeghem et al. (2012) | 19 | 0.36 | 0.26 | 0.3 |
| Chang et al. (2006) | 3 | 0.22 | 0.94 | 0.3 |
| Marble et al. (2006) | 7 | 0.07 | 0.02 | 1.8 |

Due to the zero-scaling of the field profile, all dimensions were normalised to the outer diameter of the available design volume and the application developed was as follows:

From the results in Fig. 4 and Table 1, the achieved results, beyond their novel continuous-tuning ability, outperform other reported designs/simulations in penetration depth, while having similar values of discernible thickness and field intensity/gradient, when compensating for the unavoidable distance decay. Further optimisation on the packing density was not





attempted, as limits are fabrication dependent, and the centre was left unpopulated to allow for RF coils, meaning the performance could be increased further significantly.

### 2.2.3 Optimised coupling

Beyond the applications shown, which emphasise fabrication and readjustment, the principle shown here is especially well suited for the design of strong magnets, as it near-optimally removes the degrees of freedom associated with angular alignment and mass-optimal magnet placement, which allows for development through shape/topology optimisation.

As the field of spectroscopic NMR evolves, with experiments now being done in volumes below $(5 \ nm)^3$ using NV centre spectroscopy (Staudacher et al. (2013)), superconducting magnets remain the most expensive and obtrusive element of the experimental setup. As an alternative, we set out to create the strongest possible field using permanent magnets in a geometry that allows for easy access, such as a magnet integrated into a laboratory table, and with axial symmetry, to allow for easier fabrication and assembly. Such a setup would enable a passive, low-cost magnet with an encapsulation allowing for temperature control and multidirectional access to optical instrumentation or sample feed lines.

A critical limitation of external fields is that of strong gradients, as the field decays away from the magnet. Approaches based on creating a saddle point were researched and compared to achieved solutions and found to have a limiting performance. Given the targeted application of small-scale experiments, the use of electromagnetic coils to correct for the axial gradient are far better suited considering the scaling of field gradients and power consumption at reduced scales.

As a starting point, magnetostatic reciprocity presets the optimal magnetisation direction on the half-space, a dipole field (Seleznyova et al. (2016)) centred at a normalised height of 1 above the plane and defined by its polar angle $\theta$:

$$\boldsymbol{B_d} = \frac{\mu_0}{4\pi} \frac{|\boldsymbol{m_z}|}{(\rho^2 + (z-1)^2)^{3/2}} [1.5\sin(2\theta)e_\rho + (3\cos^2(\theta) - 1)e_z] \tag{9}$$

The contour lines for the intensity of the field determined the optimal placement of the magnets to be a half-ellipsoid with a smallest axis/radius of $R_M$. This result is reminiscent of a Halbach configuration which is known to to have a logarithmic dependence on the outer radius. However, these must maintain a fully packed geometry to achieve homogeneity and thus quickly generate a demagnetised area with larger outer radii (Insinga et al. (2016a)), which limits field intensity. To overcome this, a routine was established to optimally remove voxels which would become demagnetised. As the removal of a volume sharply impacts the demagnetising field on the nearest neighbours, an iterative algorithm was necessary and thus density-based topology optimisation techniques were used, obtaining the results in Fig. 5.

The field intensity is seen to quickly attain a Halbach-like logarithmic scaling as the magnet becomes large ($R_M > 2$) compared to the normalised height, and maintains the logarithmic growth, albeit less pronounced, as demagnetised regions start being removed ($R_M > 100$). The coupling parameter explained through Eq. 3 is especially well-suited to quantitatively evaluate designs focused on intense magnetic fields. Before removal of magnetic material the results are close to unity and subsequently diminish as volumes with a large effect must be removed to avoid demagnetisation, showing the reduced efficiency of further increasing the amount of magnetic material. Values of $\eta$ above one, while generally unexpected, are caused by the demagnetising field direction matching the alignment of the magnetisation at the centre/tip of the design, constructively





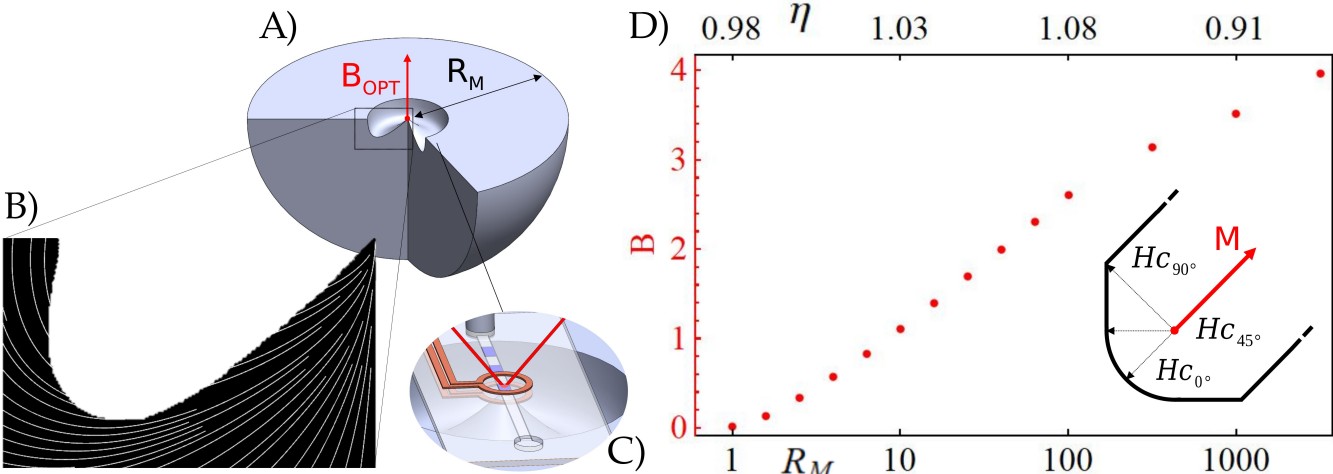

**Figure 5.** A) shows a section of the geometry (not drawn to scale) of a table-magnet that creates an optimal field at the tip of its on-axis protrusion, indicated by a red dot. B) Result of the topology optimisation for a magnet with diameter $R_M = 10^3$, with the magnetisation direction shown as white streamlines. C) An example application for microfluidic NMR. D) The dependence of the field intensity on the normalised (w.r.t. the distance above the plane being optimised for) magnet radius $R_M$. The inset right shows the phenomenological demagnetisation model for the magnetisation $M$ used during optimisation. Further detail on the model and its implementation as a regularisation are given in Section 3, in the magnetic material model and topology optimisation subsections.

magnetising that region. These results open a new avenue towards low-cost, high-performance spectroscopy and can easily be generalised, due to their scale independence, to any field requiring strong localised magnetic fields, such as needed for magnetic resonance force microscopy.

## 3 Methods

### Result normalisation

Given the scale-independence of the laws governing magnetic fields, it becomes natural to normalise all the reported values so that their use is straightforward across materials and application sizes. For this reason, all values shown are presented as adimensional. Magnetic field flux intensity is normalised per unit remanent field $B_r$, meaning field intensity will scale directly with remanence improvements, provided permeability stays constant and coercivity scales accordingly. Dimensions are shown normalised by a constant length, critical to the application, and explicitly defined. The norm of field intensity gradients is normalised by $B_r$ and multiplied by the characteristic length, to remove scale/material effects. Magnetic field inhomogeneity was defined as the relative standard deviation of the magnitude of the magnetic field, in a specific volume, and is thus shown in adimensional parts-per-million (ppm).



### FEM solver

The simulations presented were performed in a commercial FEM solver (COMSOL Multiphysics, COMSOL AB) by solving for the scalar magnetic potential, related to the magnetic field intensity as $H = -\nabla V_m$, with $\mu_0 \nabla.(H+M) = 0$. No placement, magnetisation, or angular errors were considered due to their demonstrably reducible impact. A linear constitutive relation was used, as explained in Sect. 2.1.1, and the validity of the model was checked after each simulation. Built-in routines controlling the convergence of the mesh and numerical error were used to guarantee residual errors below $1\%$ for all numerical values shown.

### Magnetic material model

NdFeB magnets, due to their high coercive strength, are often modelled (Halbach (1980)) with a constant marginal isotropic permeability ($\mu_r = 1.05$) and a constant remanent field for $M = (\mu_r - 1)H + \mu_0^{-1}B_r$. This is clear when observing simulated Halbach homogeneity profiles with radially-repeating patterns and is a crude approximation of the real non-linear anisotropic response. For this reason, and due to the difficulty in obtaining a complete model for NdFeB magnets, the values used were those reported in Katter (2005) due to their completeness: $B_r = 1.15$ T, $Hc_{0°} = 3.0$ T, $Hc_{45°} = 3.0$ T, $Hc_{90°} = 5.6$ T, $\mu_\parallel = 1.03$ and $\mu_\perp = 1.12$. These remanence and permeability values along with the directional coercivity were taken as a phenomenological model, shown in Fig. 5D and are coherent with the results shown in Martinek and Kronmüller (1990). Experimentally, one can nonetheless employ existing commercial grades of neodymium with better performance characteristics, for improved results.

### Energy minimisation

Some of the presented applications require self-alignment of arrays and thus the discovery of the equilibrium position. This is achieved in each case with a magnetostatic energy minimisation subroutine, which finds the local/global equilibrium position. This configuration is then used to compute the behaviour of the field, in each application.

In Sect. 2.2.1, computing the non-corrected discrete Halbach array, a simple routine was used to iterate the angle $\alpha$. The corrected alignment was then entered directly for comparison, but a check that a correction piece would be possible was performed beforehand.

When extrapolating to a continuous magnetisation distribution, this energy minimisation routine was performed on a scalar field, the angle of the magnetisation distribution on the available volume. The initial values were the Halbach distribution, as only a small perturbation is expected, allowing for a fast convergence. To guarantee this would be the global energy minimum, which is a non-trivial expectation because the magnet becomes much larger than the anchor, symmetry boundary conditions were removed, and the initial condition was set to be the one achieved after uniform magnetisation with a strong external field, as would be the case for a fabrication setting. The algorithm, despite not representing a physical evolution, returned a near-Halbach configuration even for the largest magnet, indicating that it would be the end configuration.



In Sect. 2.2.2, due to the rotational sectional symmetry of the design, only one of the 20 sectors was simulated, which assumes simultaneous and identical rotation of all magnets. An energy minimisation routine was used to find the equilibrium rotation angle of the cylinders when specifying the control-array at each control position.

**Topology optimisation**

In Sect. 2.2.3, density-based topology optimisation was employed through the maximisation of the objective function $f_{\mathrm{OBJ}}$ using the solid isotropic material with penalization (SIMP) method (Bendsoe and Kikuchi (1988)). The functional increased the magnetic field norm at height 1, while reducing the non-binary state of the density with the term $\rho(1-\rho)$, ensuring a sharp transition using the term $|\nabla\rho|$, and guaranteeing that the magnetic field remains within the constrains of the linear model through the 2D Heaviside function $\Theta(B - \mu_0 Hc)$, to avoid demagnetisation:

$$f_{\mathrm{OBJ}} = |B| - k_1|\nabla\rho| - k_2\rho(1-\rho) - k_3\Theta(B - \mu_0 Hc)\rho \tag{10}$$

The density was linear in the remanent magnetisation and had an initial value of 1. Several steps were used, with various size-dependent $k_i$ and mesh density, using both the globally convergent method of moving asymptotes (GCMMA in Svanberg (2002)) and sparse nonlinear optimizer (SNOPT in Gill et al. (2002)) algorithms, to allow for convergence of the density.

## 4   Conclusions

The results presented in this manuscript offer several insights into the development of permanent magnet systems for MR. By establishing a quantifiable metric for what constitutes a good magnet design, a beneficial dependence to the use of multiple magnets as well as a way to auto-align them, along with some exemplary applications of these methods, the authors hope to facilitate the future development of high performance designs, and to promote the integration of advanced numerical and experimental techniques into a now further constrained problem.

As discussed, the reciprocity principle establishes a method for developing an intensity-optimal magnet, which should be thought of as the ultimate goal of the MR magnet designer, as the field's homogeneity can then be targeted through a wealth of solutions. Despite its broad applicability and usefulness, the original principle harboured an intrinsic limitation by relying on the superposition of linear responses, which breaks down when strong non-linear effects are present. Unfortunately, these are central to state-of-the-art solutions, which nevertheless have generated impressive results up to 4 T (Kumada et al. (2001)),

by leveraging the saturation magnetisation of soft-magnetic materials of up to 2.8 T (Mehedi et al. (2017)). On the other hand, the breakdown of linearity poses severe challenges in the development of shimming systems and design trade-offs must thus be made.

The results further emphasise the need for a large filling-factor of the magnet, as this has a large effect on the magnet's coupling parameter and thus volume/cost. This approach requires a removal of in-bore passive shimming systems, which

must be placed outside, or the use of other approaches. External shimming systems could benefit from the negligible fringe field of Halbach magnets, for example, or even aid with the reduction of the demagnetisation field. Alternatively, given the

extreme homogeneity values required for spectroscopy, other approaches are likely to dominate given the lack of fabrication procedures at high enough precision, and the ability to target constant field deviations through other means. As sample sizes become smaller, the use of active shimming becomes more favourable as the power efficiency of shimming coils increases and the absolute power dissipation decreases, with an approximate dependence of $l^{-1}$ (Korvink et al. (2019)). If the magnet size remains constant and the sample/shims are down-scaled, a linear shimming profile generating the same field will span a larger gradient in the smaller enclosed volume, which introduces a further geometrical scaling factor of $l^{-1}$.

On the other hand, field inhomogeneity does not constitute a fundamental problem for NMR, as it only reduces the net measured signal, and not the local contributions. This effect comes naturally from local dephasing, which has successfully been targeted with "shimming" RF pulses (Topgaard et al. (2004)), which can periodically or continually compensate for any local deviation in phase.

Lastly, the authors targeted self-aligning arrays, which have long been an interesting phenomenon that can now be brought to MR by leveraging two of their intrinsic advantages. While self-interaction will always pose a constraint, requiring correction, a large part of the magnetostatic energy and thus the forces involved are already associated with the desired effect, through the use of an anchor magnet. This means that the forces associated with correction and assembly, even when manual, can be significantly reduced, making these two-step processes easier, as shown by the simple correction pieces employed in Fig. 2, which easily solve the limitations mentioned in Chandrana et al. (2015). Additionally, the ability to auto-align an array, appears to be the only way to break free from deviations caused by the use of a limited number of magnets, which in turn limit the overall repeatability of designs, leaving them susceptible only to assembly/scaffold-fabrication offsets.

*Author contributions.* PS and JGK developed the concept. PS developed the theory, and conducted the simulations. PS, MJ and JGK wrote the paper.

*Competing interests.* The authors declare that they have no conflicts of interest.

*Acknowledgements.* The authors would like to thank Dr. Erwin Fuhrer, Dr. Neil MacKinnon, and Prof. Yongbo Deng for helpful and fruitful discussions. PFS acknowledges Bürkert GmbH & Co. KG for partial funding for this project. JGK and MJ acknowledge the DFG for partial funding [contracts KO 1883/29-1 and KO 1883/34-1].



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
