# Peer review of "Magnetostatic reciprocity for MR magnet design"

_Magnetic Resonance, 2021_

## Community Comment (CC3)

Comments on Jean's reciprocity.  Tom Barbara 5/13/21

Here are further aspects of my previous comments on the Jeans reciprocity that this submission uses in their design efforts.  For each *prescribed* source $\mathbf{M}_1$ and $\mathbf{M}_2$ we have (in Gaussian units)

$$\boldsymbol{B}_i - \mathbf{H}_i = 4\pi\,\mathbf{M}_i \quad (1)$$

$$\boldsymbol{\nabla} \times \boldsymbol{H}_i = 0$$

$$\boldsymbol{\nabla} \cdot \boldsymbol{B}_i = 0$$

It is important to emphasize that the $\mathbf{M}_i$ are prescribed sources, that is that the magnetizations are saturated or fixed, as discussed in Jackson's book on E&M.

We can take scalar products of (1) for each source and $\mathbf{H}_j$ for the other, and take their difference

$$\mathbf{H}_2 \cdot (\boldsymbol{B}_1 - \mathbf{H}_1) = 4\pi\,\mathbf{H}_2 \cdot \mathbf{M}_1$$

$$\mathbf{H}_1 \cdot (\boldsymbol{B}_2 - \mathbf{H}_2) = 4\pi\,\mathbf{H}_1 \cdot \mathbf{M}_2$$

The terms in **H** for each source are thereby eliminated.  After integrating over all space, the mixed field terms vanish by the fact that they are "Helmholtz pairs", so that we are left with

$$\int d^3x\ \mathbf{H}_1 \cdot \mathbf{M}_2 \;=\; \int d^3x\ \mathbf{H}_2 \cdot \mathbf{M}_1$$

I believe this is a clearer statement of the Jeans reciprocity.  Of course the validity of this theorem does require linear superposition which follows from the assumption that the sources are prescribed.

---

## Author Comment (AC1)

Karlsruhe Institute of Technology

KIT | IMT | Postfach 3640 | 76021 Karlsruhe, Germany

Dr. Robert Tycko
Associate Editor
Magnetic Resonance

**Institute of Microstructure Technology**

Head:   Prof. Dr. Jan G. Korvink

Hermann-von-Helmholtz-Platz 1
76344 Eggenstein-Leopoldshafen, Germany

Phone:  +49 721 608-29314
Fax:    +49 721 608-22740
Email:  jan.korvink@kit.edu
Web:    https://www.imt.kit.edu

Official in charge:   Jan Korvink
Date:    June 23, 2021

[Figure]

[Figure]

**Revision, Manuscript mr-2021-34**

Dear Dr. Tycko,
Dear Rob,

We thank you for considering our manuscript entitled **Magnetostatic reciprocity for MR magnet design** as a revision to **Magnetic Resonance**. We highly appreciate the time you and the reviewers have invested in our manuscript and are grateful for the valuable feedback.

We have used the opportunity to revise our manuscript considering the comments and remarks raised by the reviewers. Below is our point-by-point response to the questions that were raised, with our responses written in blue. Also, a copy of the modified manuscript is attached where the modifications are highlighted. Taking the comments into consideration, we strongly believe the quality and clarity of the manuscript have been improved.

**Reviewer comments 1**

- **Reviewer 1**: Terms like: anchor magnet, anchor volume, $B_{\text{target}}$, minimum discernible thickness, etc, are used without definition. For a reader who is not intimately involved in this kind of design work, clear definitions when these terms are used would help.
  We have attempted to better explain the missing concepts as detailed below.
  An **anchor magnet** is termporarily placed in the volume at a position where a specific target field should be generated (i.e. where the sample will eventually be placed) and is used to align the other small magnets that will actually be used to produce the target field after the anchor is removed. The introduction of the energy reciprocity now introduces the concept better.
  The **anchor volume** is the volume taken up by the anchor magnet (i.e. eventually, when the anchor is removed, this will be the sample volume). This is better elucidated in the text now.
  $B_{\text{target}}$ is the target field that will remain at the site of the sample volume (also the anchor volume) by the remaining small magnets after removing the anchor magnet.
  The **minimum discernible thickness** is defined on line 196 and is now further expanded for clarity.

- **Reviewer 1**: In section 2.1.1 it would help to refer to Figure 1, and to state clearly what is being contemplated. The way this material is framed, there is a vector potential A, but no words are said about how A is created. It is then a confusing surprise to see BrA which is apparently a remnant field associated with magnetic material that is associated with A. It might be much simpler to frame the problem more symmetrically from the start, perhaps labelling the domains with more descriptive names such as 'sample' and 'external,' or even just 1 and 2. Each of these regions could contain magnetic material and produce fields in the other domain? The 'anchor' could then be defined as a magnet in region 1 for design and/or fabrication purposes and 2 will contain the

**Karlsruhe Institute of Technology (KIT)**
**Kaiserstraße 12**
**76131 Karlsruhe**

**President: Prof. Dr.-Ing. Holger Hanselka**
**Vice Presidents: Michael Ganß, Prof. Dr. Thomas Hirth**
**Prof. Dr. Oliver Kraft, Christine von Vangerow,**
**Prof. Dr. Alexander Wanner**

**Bundesbank Karlsruhe**
**BLZ 660 000 00 | Kto. 6600 1535**
**BIC/SWIFT: MARK DE F1660**
**IBAN: DE07 6600 0000 0066 0015 35**
**USt-IdNr. DE266749428**

[Figure]

Karlsruhe Institute of Technology

finished designed magnet. It is unclear to me why the vector potential is introduced at all here. It is never used, except as a label for one of the domains.

The authors fully agree. The use of the vector potential A was fully removed from the manuscript to avoid confusion with the anchor magnet A. Section 2.1.1 now has a clear text reference as to which magnet items correspond to the magnet under development, and which correspond to a support anchor.

- **Reviewer 1**: On line 81, the introduction of $B_\text{target}$ is unclear. What is $B_\text{target}$? Can the origin of the minus sign be explained simply?

  As this is addressed above, we believe the meaning should now be clear.

- **Reviewer 1**: Line 151: 'zero scaling law' What exactly is meant here?

  'Zero scaling law' refers to the scale invariance of magnetic field profiles when normalized by a characteristic length, as is done in the paper (i.e. a magnet that is twice larger in every direction will generate a magnetic field that is twice "larger" in space). Terminology adapted in the paper.

- **Reviewer 1**: Fig 4A: the coloured figure has many colours in it that aren't in the scale bar?

  The scale bar in 4A was reduced to best show the range of magnetic field strength at the penetration depths depicted in 4B. As 4B shows the minimum discernible depth and two contours, for simplicity, 4A attempts to give the reader an idea of the intensity profile in the same volume.

- **Reviewer 1**: What are the different panels in Fig 4B? What are the x- and y-axes of these panels?

  The panels show the minimum discernible thickness profiles as the magnet is adjusted. The x- and y-axes are dimensions normalized to the characteristic length defined for the profilometry magnet as discussed in the 'methods' section. This is now made clear in the legend of the image. The authors apologize for the confusion.

- **Reviewer 1**: Please state clearly how the control actuation works? There is a hint in Figure 4 that they are moved up and down, though without any sense of scale it is unclear how far. How much force is required to move them and hold them in position?

  The legend of 4A now shows the actuation range of -0.06 to 0.08. The force applied is left undiscussed due to its lack of scale invariance and the ability to generate sufficiently large forces along any direction using, for example, a screw.

- **Reviewer 1**: It might help to clearly define the function and requirements of a profiling magnet.

  Profiling magnets were now expanded upon, aiming for an easier understanding by readers unfamiliar with their application.

- **Reviewer 1**: Please explain the origin of Eq 8 more clearly and ensure that all quantities in it are clearly defined.

  The authors agree. The origin of the equation is now made clear in the text, right before its introduction, and in the introduction to the profiling magnets.

- **Reviewer 1**: What is the $d_\text{length}$ scale of T? I found: Due to the zero-scaling of the field profile, all dimensions were normalised to the outer diameter of the available design volume. Maybe a scale bar could go on Figure 4A?

  T varies from $10^{-3}$ to $10^{-2}$ relative to the normalizing dimension, which is the outer diameter of the magnet.

- **Reviewer 1**: I was unable to interpret Table 1. What is $d$?

  The authors apologize for the lack of clarity. $d$ is the normalized penetration depth. All variables are now explained in the caption of the table.

- **Reviewer 1**: line 216-217: "found to have a limiting performance" is very vague. Could you be more specific?

  The 'limiting performance' mentioned is now rephrased as a significantly reduced $B$ field strength when compared to approaches not using a saddle point.

- **Reviewer 1**: Fig 5: The upper x-axis ($\eta$) is non-monotonic? Evenly spaced tick-marks imply some quantity that is varying smoothly and uniformly. The $\eta$ axis is very misleading. It would be helpful to remind the reader what $\eta$ is.

  $\eta$ is now reviewed in the caption of the image. $\eta$ is the numerically computed coupling factor, which is a function of the dimension in the x-axis, the dimension of the magnet.

- **Reviewer 1**: Fig 5D inset. I don't understand what is being drawn at all.

  The inset shows a standard representation of the phenomenological model for the demagnetization of a volume.

[Figure]

Karlsruhe Institute of Technology

It shows what field is expected to move the 'frozen' magnetization from its state. The maximum value of the well-known coercive field, $H_c$, changes according to the direction it is applied. The maximum field applied at an angle of 180 degrees relative to the magnetization is defined as $H_{c0}$ and it varies according to the model shown (e.g. same norm for a 45 degree offset).

- **Reviewer 1**: As a non-expert in the work being done here, I found the presentation of the third application to be completely opaque. Only with some difficulty could I even follow the basic idea of what the goal was, and even so was left wondering at the end what the phrase: "unbound magnetic field strength" (line 54) meant. Is the goal simply to maximize the field strength in that case?
  Yes, that is correct. Simply put, the goal is to find a way to continually increase the field without the magnet demagnetizing itself. This is done iteratively as explained in the text.

- **Reviewer 1**: Could the authors comment on how the structure in Figure 5 might be fabricated? The abstract does emphasize manufacturability.
  The manufacturing of the first two structures are discussed, and involve either cleanroom processing, or normal workshop assembly. The authors definitely find the realisation of the third structure a very interesting follow-up to the theoretical work shown. However, its fabrication would be exceedingly complex to discuss in the manuscript, requiring numerical modeling/optimization of a non-continuous magnetization (i.e., blocks uniformly magnetized). This would have to become scale-dependent (as different scales require different techniques) and have a drastically scale-dependent end result, and hence was not included. We hope that this is acceptable.

- **Reviewer 1**: I think it would help to mention the non-dimensionality of many of the variables much earlier in the paper, before it might cause confusion.
  The authors agree. The method section is now introduced before the results section and some adjustments were made for clarity.

- **Reviewer 1**: Figure 2C is unclear. How do the upper and lower x axes relate to each other? Why does the upper axis increase nonlinearly? Why is the region $1 < d_M < 1.5$ the interesting region? Isn't $d_M < 1$ the region of interest?
  The methodology is the same as applied to $\eta$, and mentioned above. The angle $\alpha$ is numerically computed from the independent variable $d_M$. $d_M$ is defined as the outer diameter of the structure. The magnets generating the field have a size $1 - d_M$ and therefore go to zero size as $d_M$ approaches 1, which leads to the 0 field shown for $d_M = 0$.

- **Reviewer 1**: The caption to Figure 3 suggests that the structures under consideration are those of Fig 2, though Fig 3A appears to simulate an annular ring of magnetic powder rather than the discrete cylinders of Figure 2. 20) It would be helpful to give some sense of the scale of the calculations. Are these things that could be done in a few minutes on a desktop computer, or are they hours on a large cluster?
  The authors agree. In the introduction of the numerical method it is now stated that the methods implemented can be computed in a laptop in minutes to hours (depending on the application).

- **Reviewer 1**: Many additional details could be provided. I do not think enough information is provided in the methods that these results could be reproduced.
  Several more details were added to the text according to your helpful contributions and those of the community and other reviewers. The authors request your understanding that, if complex applications within the field of MRI had been chosen to best showcase the impact of the theory, the complete explanation of all these sub-fields would have been untenable for a single manuscript.

**Reviewer comment 2**

- **Reviewer 2**: The paper begins with a description of the reciprocity principle for magnetostatics, and then moves through several examples where the reciprocity principle is used to analyze and/or optimize the performance of an array of permanent magnets. However, as far as I could follow, the manuscript lacked clear and adequate descriptions of the problems they were trying to solve. Part of the confusion stems from the fact that the paper is full of jargon unfamiliar to non-experts which is never defined. I recommend the authors rework the manuscript with an eye towards simplifying the presentation and clarifying the problems they are trying to address. Some particular points of clarification:

[Figure]

Karlsruhe Institute of Technology

We thank the reviewer for this observation. We have carefully reworked the paper, also heeding all the reviewer comments.

- **Reviewer 2**: For non-experts, it would be helpful to see some dimensionful units. What are the units of $B$, for example? How physically large are these arrays? At the end of the manuscript it is explained that all quantities presented are dimensionless, however this should be made clear from the beginning.
  The authors agree with all the reviewers and the methods section was now moved to be read before entering the discussion of the numerical results. Given the scale invariance of the magnetic fields being researched, these arrays can potentially have any size. A reference example for $B_r$ (while being fully material dependent value) is introduced in the manuscript as 1.15 T for NdFeB. As such, all $B$ fields shown are simply $B * 1.15$ T when not normalized.

- **Reviewer 2**: It is not clear to me how Equation 2 as written follows from Equation 1, if $M$, $H$, and $B$ are defined as described in the preceding paragraph. Where do the self-interaction terms $H_a * H_a$ and $H_m * H_m$ come from?
  Equation 2 is obtained by introducing the definition of $M$ stated in the line above the Eq 2, into equation 1, and rearranging the terms for simplicity. The self-interaction terms appear directly from this substitution. The interaction volume also went from $R^3$ to $V_M$ and $V_A$, and this is now made explicit in the text too.

- **Reviewer 2**: Figure 1: The term "anchor magnet" is never clearly defined in the text.
  The authors agree, and the text was adapted to make the use of the term easier to understand.

- **Reviewer 2**: Figure 2:
  - **Reviewer 2**: What is $d_m$? It seems either it is not clearly defined, or it is not used with a consistent definition.
    $d_M$ is the size of the outer radius of the structure displayed. This is normalized to the inner diameter of the structure, which is usually the limiting volume (i.e. the target volume where the field is to be created). This is now made clear in the caption of Fig 2, beyond its previous definition in lines 146 and 147.

  - **Reviewer 2**: Again, for non-experts, it would be helpful to see some dimensionful units here - what is the rough size of the $B$-field that can be produced by this microarray, for example?
    The normalized value of $B$ goes, in the graph, from 0 to 0.5 $B_r$. This corresponds to 0 to 0.575 T when using the NdFeB ($B_r = 1.15$ T) material mentioned above.

  - **Reviewer 2**: What are the colors in Figure 2A) indicating?
    The colors of Fig 2A are mostly illustrative, but represent the near-homogeneous field intensity, in the target volume (red circular volume), created by a Halbach magnet arrangement, as predicted by the FEM simulation.

  - **Reviewer 2**: As far as I can tell, $B$ and $C$ both show different aspects of the design assembled with and without the outer alignment structure, but the caption seems to indicate $B$ and $C$ deal with the two respective cases?
    The interpretation is absolutely correct. The caption was rephrased for better clarity.

  - **Reviewer 2**: 2C is quite confusing to me. Are the $d_m$ and $\alpha$ axes correlated? What exactly is being plotted here?
    $d_M$ is the independent variable and $\alpha$ is computed for varying values of $d_M$, just like $B$ or $\sigma_B$, and therefore the numerical values plotted are $\alpha(d_M)$

- **Reviewer 2**: Figure 3:
  - **Reviewer 2**: How is this self-alignment achieved?
    The procedure is the same as for Fig 2, an anchor is used and the structures/powder are allowed to rotate under the influence of the total magnetic field they experience. This comparison to fig 2 is now made more explicit in the text.

  - **Reviewer 2**: Is there an intuitive reason why $\sigma_B$ shows the structure it does as a function of $d_M$? It is not clear to me what is being plotted here, what is the relationship between the magnet diameter and the anchor?
    As mentioned in line 165, field intensity levels-off as the effect of the anchor (the desired alignment) starts

[Figure]

Karlsruhe Institute of Technology

being dominated by the effect of neighbouring powder materials. However, the authors do not have an intuitive explanation as to why this leads to the dependence found.

- **Reviewer 2**: Figure 3:
  - **Reviewer 2**: A clearer presentation of the toroidal array would be helpful.
    Several representations of the array were attempted and the one obtained with a section removed was chosen for compactness and clarity. The caption of the image is now adapted to make this more clear to the reader.
  - **Reviewer 2**: What is 4B showing? The caption is unclear, 4B is identified twice. Which plots show what?
    The authors agree. The caption is now adapted to correct this and give better insight.
  - **Reviewer 2**: How large are the T values presented? Again, some feeling for dimensions would be extremely helpful.
    Due to the scale invariance of the problem, dimensions can vary greatly. If the outer diameter of the magnet were 100 mm, T would range from 0.1 to 1mm (shown normalized as $10^{-3}$ to $10^{-2}$).
- **Reviewer 2**: Figure 5: What is 5D showing? What does the red represent? Why is the $\eta$ axis non-monotonic?
  In 5D, the normalized magnetic fields obtained are shown in red. As used before, $\eta$ is shown as a function of $R_M$ and therefore it is possible that $\eta(R_M)$ is non-monotonic.
- **Reviewer 2**: There are several different notations used throughout the manuscript. Equations 1 and 2 use one vector notation, Equation 3 uses a second, different notation, Equations 4 and 5 use a third notation, and Equation 9 uses a forth notation.
  The authors fully agree. The equations were adapted to strictly adhere to the same notation.

**Reviewer comments 3**

- **Reviewer 3**: "This simplifies Eq. 1 to:" But Eq. 2 does not look like a simplification, only if you know exactly what you are looking for. Perhaps rather this should be called a transformation?
  The authors agree. The text was adapted for better clarity.
- **Reviewer 3**: "Whereas this approach matches exactly the ideal coupling between two equal magnets," it is not clear what coupling between magnets means here
  The authors fully agree. The sentence is now better explained. In simple terms, the 'desired' results would be that only the first two terms of the equation would exist, at which point a formally correct superposition principle would be possible. Instead, the next sentence shows why and how an approximation is reasonable.
- **Reviewer 3**: "Assuming $B_{\text{target}} = -B_r A$ in Eq. 1, one can see how maximum compliance with the target field is given by the minimum magnetostatic energy condition." Target field is not defined, and it is not clear where the minimum magnetostatic energy conditions comes from or what it refers to.
  The authors agree. $B_{\text{target}}$ is now better defined in the text. The compliance and minimum energy conditions are also better explained.
- **Reviewer 3**: Section heading "High-n applications" should probably be reworded to simply say something like "Arrangement of n magnets" (does it matter what kinds of magnets?)
  Agreed. The suggestion was directly implemented by changing the title of the section. The results consider the scaling of multiple metrics with the number of magnets, and therefore the type of magnets (i.e. material or shape) should not be relevant as long as the assumptions used still hold.
- **Reviewer 3**: "which takes small results in neodymium magnets as $\mu_k = 1.03$ and $\mu_k = 1.12$" what does small results mean?
  The term being discussed is being multiplied by $(\mu - 1)$, which takes a value significantly below 1 due to the use of NdFeB, unlike what would happen for other ferromagnetic materials ($\mu \gg 1$). The text was adjusted for clarity.
- **Reviewer 3**: "Similarly, one can deduce a reciprocity between two regions in space and the magnets/fields contained therein." it is not clear whether this sentence indeed says what the authors wish to say.
  The authors aimed to make a logical comparison between the reciprocity being discussed in the manuscript and that which was published by Hoult on electromagnetic induction in a coil.

[Figure]

Karlsruhe Institute of Technology

- **Reviewer 3**: Convoluted sentence: "One can nonetheless estimate the relative inefficiency of a design in optimally placing magnetic energy in the anchor volume in which a sample will be placed."
  The authors agree. The sentence was rephrased for clarity.

- **Reviewer 3**: "intensity-optimal magnet, which should be thought of as the ultimate goal of the MR magnet designer, as the field's homogeneity can then be targeted through a wealth of solutions." What is an intensity-optimal magnet and how do you balance this goal against the goal of homogeneity?
  Intensity optimality refers to using a certain mass/volume optimally to create the strongest magnetic field possible, as introduced in the text. "How to balance strength with homogeneity", while quite relevant as an academic question, is somewhat outside the scope of the paper, as it cannot be answered directly through the reciprocity principle. Furthermore, the need for homogeneity is drastically different depending on the MR application, ranging from strong NMR linewidth requirements, to far relaxed EPR linewidths, to a useful inhomogeneity (i.e. gradients) when performing MRI experiments. This further takes the discussion outside of the scope of the manuscript.

- **Reviewer 3**: "On the other hand, field inhomogeneity does not constitute a fundamental problem for NMR, as it only reduces the net measured signal, and not the local contributions. This effect comes naturally from local dephasing, which has successfully been targeted with 'shimming' RF pulses (Topgaard et al. (2004)), and which can periodically or continually compensate for any local deviation in phase." What type of phase is refered to?
  The phase mentioned is the MR signal's phase which, when different across the sample/imaging-pixel, contributes to T2* decay. Formally it can be defined as $arg(B_r \cdot M)$, where $B_r$ is the receptive field of the Rx coil and $M$ the magnetization state.

We appreciate your consideration of our manuscript and look forward to hearing from you.

On behalf of all the authors
Yours Sincerely,

Jan Korvink
Institute of Microstructure Technology
Karlsruhe Institute of Technology